# Effect of Water Area and Waterweed Coverage on the Growth of Pond-Reared *Eriocheir sinensis*

**Yongcheng Tang [1,†], Jiao Peng [1,†], Jiahao Chen [2], Yunlin Zhao [1], Yi Ding [1], Jingyi Dai [1], Zhiyuan Hu [3], Tian Huang [4], Meng Dong [3] and Zhenggang Xu [1,2,4,*]**

1    Key Laboratory of Forestry Remote Sensing Based Big Data & Ecological Security for Hunan Province, Central South University of Forestry and Technology, Changsha 410004, China
2    College of Forestry, Northwest A & F University, Yangling 712100, China
3    School of Materials and Chemical Engineering, Hunan City University, Yiyang 413000, China
4    Engineering Research Center for Internet of Animals, Changsha 410128, China
*    Correspondence: xuzhenggang@nwafu.edu.cn; Tel.: +86-18684945647
†    These authors contributed equally to this work.

**Abstract:** Water area and waterweed coverage are the key environmental factors for ecological breeding of *Eriocheir sinensis* in ponds. In order to explore the effects of above two factors on the growth of *E. sinensis*, three groups of experiments were set up: low coverage small area (C1S1), high coverage small area (C2S1), and high coverage large area (C2S2), and water environmental factors and the growth of *E. sinensis* were monitored. The results showed that the dissolved oxygen of ponds with different waterweed coverage was significantly different ($p < 0.05$), and the phosphate in ponds changed significantly from July to October ($p < 0.05$). The correlation analysis showed that $NH_3$-N and pH were significantly positively correlated ($p < 0.05$). At the same time, there was a significant negative correlation between $NH_3$-N and DO, $H_3PO_4$ and pH ($p < 0.05$). Further analysis of the relationship between surface area, waterweed coverage and environmental factors revealed that the surface area and waterweed coverage were closely related to $NH_3$-N, $H_3PO_4$ and DO. From July to October, the differences in morphology and weight of *E. sinensis* in different ponds became more and more significant. In terms of growth rate, C1S1 with a small area and low coverage had a downward trend, while C2S1 and C2S2 were the opposite. In the final stage of the experiment, C2S2 had the highest yield (0.1311 kg/m$^2$), and C1S1 had the lowest yield (0.0600 kg/m$^2$). Then, the ponds with high waterweed coverage and large area can bring better benefits.

**Keywords:** *Eriocheir sinensis*; water area; waterweed coverage; water quality; crab yield

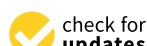

## 1. Introduction

Aquatic products are rich in protein, eicosapentaenoic acid (EPA) and docosahexaenoic acid (DHA), which are indispensable nutrition for human brain development, intelligence development, improvement of human immune function and health quality source [1]. In recent years, aquaculture had developed rapidly, and artificial aquaculture accounted for more than half of the world's aquatic products [2]. China is the largest aquaculture country in the world, with rich fishery resources and a long history of aquaculture [3]. In 2019, the total amount of aquatic products in China was 64.8 million tons, and the ratio of aquaculture output to fishing output was 78.4:21.6 [4]. Chinese mitten crab (*Eriocheir sinensis*) is an important aquatic product, and many Asian countries have a tradition of eating hairy crabs, especially in China, although the animals may pose a risk of biological invasion in Europe and America. There are records indicating the raising and feeding Chinese mitten crabs in China as early as 1844 [5]. In recent years, Chinese mitten crab culture has developed rapidly. Generally speaking, Chinese mitten crabs would be cultured for two years until they were put on the market [6]. In the first year, juvenile crabs are cultivated to a button size and then to a marketable size with sexual maturity in the next

year [7]. After putting the juvenile crabs into the water, germplasm and specification are determined. The growth environment is particularly important. Cage culture and pond culture are the two main methods of crab culture at present. In the wild water, cage culture are employed and crabs are usually stocked at low density because of no supplementary feed, which limited the accurate control of the breeding environment [8,9]. At the same time, cage breeding in wild water would also pollute the water and cause eutrophication [10]. In order to further protect the water environment and increase production, the pond culture of crabs is constantly replacing cage culture in the wild. Consequently, since 2006, pond culture has dominated crab farming, not only in terms of breeding areas, but also in terms of output. Pond farming is also constantly transitioning to ecological farming in order to provide people with more ecologically healthy aquatic products.

The output and quality of aquatic products are closely related to the environment. Waterweed coverage was one of the most important ecological factors in Chinese mitten crab culture [11]. Waterweed could not only eliminate the content of organic matter, ammonia nitrogen, nitrite and other substances in the water, but also provided crabs with green feeds, multivitamins and appropriate habitats [12]. *Elodea canadensis* is the most common aquatic plant in the pond. Previous research has shown that *E. canadensis* enhances growth and improves the nutritional quality of *E. sinensis* [13]. At the same time, culture density is another important factor in the culture of *E. sinensis.* In order to obtain a high economic benefit, it was commonly used to increase the yield per unit by raising the stocking density [14]. However, the high density of crabs leads to internal competition for space and bait. Thereafter, in order to adapt to a high-density environment, the energy consumption of crabs increased, resulting in a decrease in mean weight, feed conversion efficiency and survival rate [8]. The stocking density was determined by the number of crabs and the surface area [11]. Under the same stocking density, to the *E. sinensis* in pond requires maintaining a certain depth of water, as each crab living space is not consistent in the ponds limited by different surface areas.

In order to obtain better benefits, screening out the best growth environment for crabs is needed. In this study, we compared the growth of *E. sinensis* in different waterweed coverage and different surface areas, and identified the best waterweed coverage and surface area for *E. sinensis* farming. The aim of this work is to provide a theoretical basis for the successful cultivation of *E. sinensis*.

## 2. Materials and Methods

### 2.1. Experimental Design

The study was carried out in the Dongting Lake area, which is an important breeding area for Chinese mitten crabs, from July to October 2016. In March 2016, ponds were prepared prior to stocking and the preparation included pond drainage, air drying for 20 to 30 days, and the application of 500 to 1000 kg/hm$^2$ of lime (calcium oxide, CaO). The water was then irrigated to a depth of 30 to 50 cm, and macrophytes (mainly *E. canadensis*) seedlings were transplanted (500 to 1000 kg/hm$^2$). The stocking density ranged from 16,000 to 18,000 crabs ha$^{-1}$.

The experimental plots were elevated dikes of around 1 m in height. Each plot had an outlet/inlet connected to two central irrigation channels (Figure 1). Primarily to improve water quality and increase dissolved oxygen, the water exchange was managed independently for each pond. Fifteen to thirty percent water of each pond was exchanged irregularly every 3 to 5 days. The quantity of inflow was generally greater than the outflow to offset evaporation and leakage. In addition, meshed nylon nets were set up around the pond for keeping crabs and for preventing indigenous fish and aquatic predators outside. The feed consisted of one serving per day of fish, wheat, soybeans and corn (at 5:00 p.m.). No chemical pesticides were applied in order to ensure product safety.

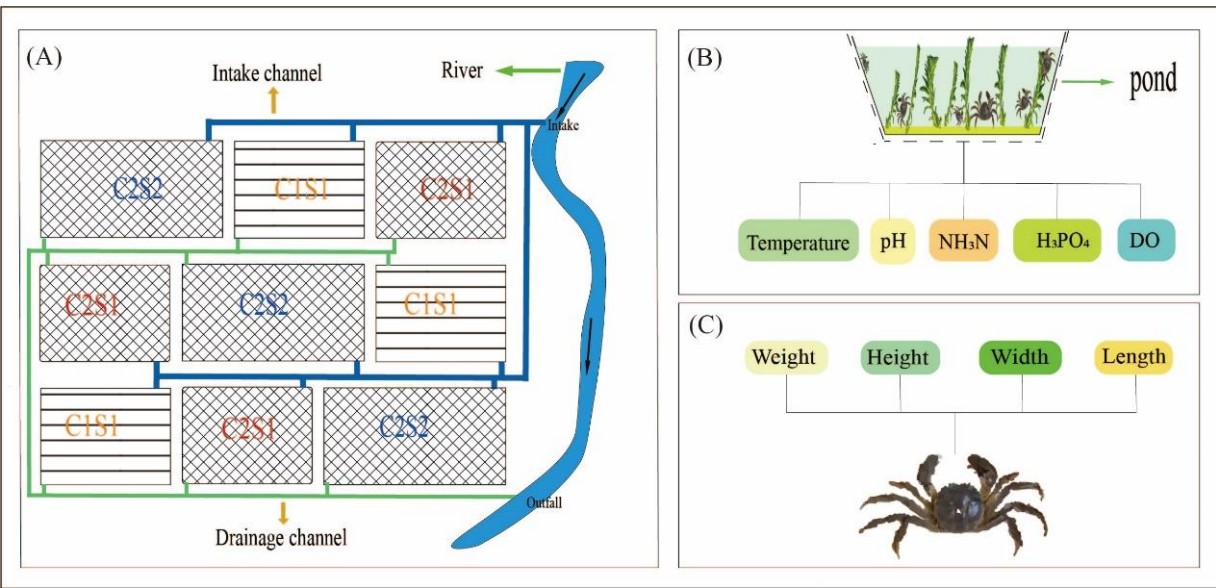

**Figure 1.** Schematic diagram of experimental design (**A**), quality indicators of water environmental (**B**) and indicators of *Eriocheir sinensis* (**C**).

The experiment included three treatments. Each treatment had three replicates. The three treatments were as follows: the low percentage of waterweed coverage with 50% and small surface area with 0.75 ha (C1S1), the high percentage of waterweed coverage with 70% and small surface area with 0.75 ha (C2S1), and the high percentage of waterweed coverage with 70% and the large surface area with 1.5 ha (C2S2). The previous experience of farmers represented that the survival rate and the size of crabs were low under the condition of a low percentage of waterweed coverage and large surface area, so this treatment was not included in our experiment (Figure 1A). The initial weight of each crab larvae was about 0.135 ± 0.025 g.

### 2.2. Monitoring of the Water Characteristics

Water samples were taken from the different treatments from the 11th to the 20th of each month, and the measurements were conducted from July to October until the Chinese mitten crabs matured. We investigated water quality parameters including dissolved oxygen (DO), temperature (T) and pH. During the daytime, the above parameters were monitored using electronic probes and a portable multi-parameter instrument (Shanghai Leici JPB-607A), beginning at 8:30 a.m. and samples were taken every two hours. Ammonium-nitrogen ($NH_3$-N) and orthophosphate ($H_3PO_4$) were determined at 8:30 a.m., 12:30 p.m. and 6:30 p.m., respectively (Figure 1B). Water samples were collected from each experimental pond and analyzed by a water quality analyzer (Octadem W-II).

### 2.3. Data Collection for the E. sinensis

From July to October 2016, we compared the morphological characteristics of *E. sinensis* between the three treatments. Ten crabs (five males and five females) were randomly selected from each pond on the 20th of each month for measurement. The crabs were weighed and measured for carapace length, carapace width and carapace height. A wet towel was used to wipe off the surface water of crabs before the measuring (Figure 1C). The growth rate was expressed as specific growth rates (SGR) based on the following formula:

$$SGR = [(\ln(Wf) - \ln(Wi)) \times 100]/T \tag{1}$$

where Wf is the final weight (g), Wi is the initial weight (g) and T is the experiment time (days).

In October 2016, except for measuring the morphological characteristics of *E. sinensis*, we also analyzed the quality characteristics. The wet weight of males and females were similar, and the gonads of adult crabs were mature at this time. The gonad, hepatopancreas and all somatic muscle of each crab were dissected, weighed and stored for analysis. We then further examined and compared tissue indices, such as the condition factor (CF), the gonadosomatic index (GSI), the hepatosomatic index (HSI), and the muscular index (MI) from three different culture ponds. The CF, GSI, HSI, MI and total edible content was calculated according to the formula [15,16]:

$$CF = Wb/L \times 100 \qquad (2)$$

$$GSI\ (\%) = Wg/Wb \times 100 \qquad (3)$$

$$HSI\ (\%) = Wh/Wb \times 100 \qquad (4)$$

$$MI\ (\%) = Wa/Wb \times 100 \qquad (5)$$

$$Total\ edible\ content\ (\%) = GSI + HSI + MI \qquad (6)$$

where Wb is the body weight (g), L is carapace length, Wg is the gonad weight (g), Wh is the hepatopancreas weight (g), and Wa is the all somatic muscle weight (g).

### 2.4. Statistical Analysis

Results were presented as mean ± S.D. with three replicates. Data were subjected to a normality test, one-way ANOVA and LSD multiple range test using SPSS 26 software. Principal component analysis (PCA) was conducted using Canoco 5.0 for surface area and waterweed coverage to identify the main factors of the water quality. The difference was considered significant at $p < 0.05$. Origin 2021 was used for plotting.

## 3. Results

### 3.1. Fluctuation Characteristic of Water Environment Quality

3.1.1. pH Change of Pond Water

The culture water of *E. sinensis* was weakly alkaline (Table 1). During the experiment period, the fluctuation of pH was not manifest. The maximum value of the diurnal variation was 1.85, and the minimum value was just 0.1. Furthermore, there was no significant difference in pH among the three treatments ($p > 0.05$). Probably due to a large amount of feed input, the water pH of each pond had a certain upward trend from July to September, but it returned to the original level in October when the breeding was basically completed.

**Table 1.** Monthly variation of water pH in different ponds.

| Month | Time | S1C1 | S1C2 | S2C2 | *p* Value |
|---|---|---|---|---|---|
| July | 08:30 | 7.456 ± 0.289 | 7.470 ± 0.363 | 7.421 ± 0.240 | 0.940 |
| | 12:30 | 7.326 ± 0.319 | 7.489 ± 0.277 | 7.423 ± 0.174 | 0.430 |
| | 18:30 | 7.540 ± 0.347 | 7.626 ± 0.273 | 7.438 ± 0.266 | 0.422 |
| August | 08:30 | 7.488 ± 0.299 | 7.560 ± 0.225 | 7.493 ± 0.325 | 0.822 |
| | 12:30 | 7.546 ± 0.331 | 7.577 ± 0.287 | 7.523 ± 0.273 | 0.929 |
| | 18:30 | 7.606 ± 0.331 | 7.632 ± 0.236 | 7.556 ± 0.212 | 0.810 |
| September | 08:30 | 7.519 ± 0.397 | 7.461 ± 0.371 | 7.748 ± 0.467 | 0.349 |
| | 12:30 | 7.743 ± 0.607 | 8.090 ± 0.545 | 7.949 ± 0.520 | 0.470 |
| | 18:30 | 8.154 ± 0.338 | 8.174 ± 0.546 | 8.201 ± 0.373 | 0.973 |
| October | 08:30 | 6.940 ± 0.263 | 7.140 ± 0.196 | 7.112 ± 0.247 | 0.144 |
| | 12:30 | 7.140 ± 0.151 | 7.130 ± 0.183 | 7.240 ± 0.337 | 0.528 |
| | 18:30 | 7.433 ± 0.403 | 7.344 ± 0.505 | 7.350 ± 0.417 | 0.892 |

### 3.1.2. Water Temperature Change

According to the daily fluctuation of temperature values measured from July to October, all temperatures reached a peak at about 12:30 p.m. (Figure 2). The average water temperature in July and August was above 30 °C, but not in September and October. During the monitoring period, the water temperature of Pond C2S1 and Pond C2S2 with high waterweed coverage did not exceed 38 °C, which was considered to be an important critical temperature [17]. The highest water temperature of Pond C1S1 with low waterweed coverage reached 38.9 °C in August. The water temperature of Pond C2S1 was higher than that of Pond C1S1 most of the time, and the water temperature of Pond C1S1 was higher than that of Pond C2S1 in August. From morning (8:30 a.m.), noon (12:30 p.m.) to evening (6:30 p.m.), statistical analysis showed that there was no significant difference in the water temperature between C1S1, C2S1 and C2S2 at the same time ($p > 0.05$).

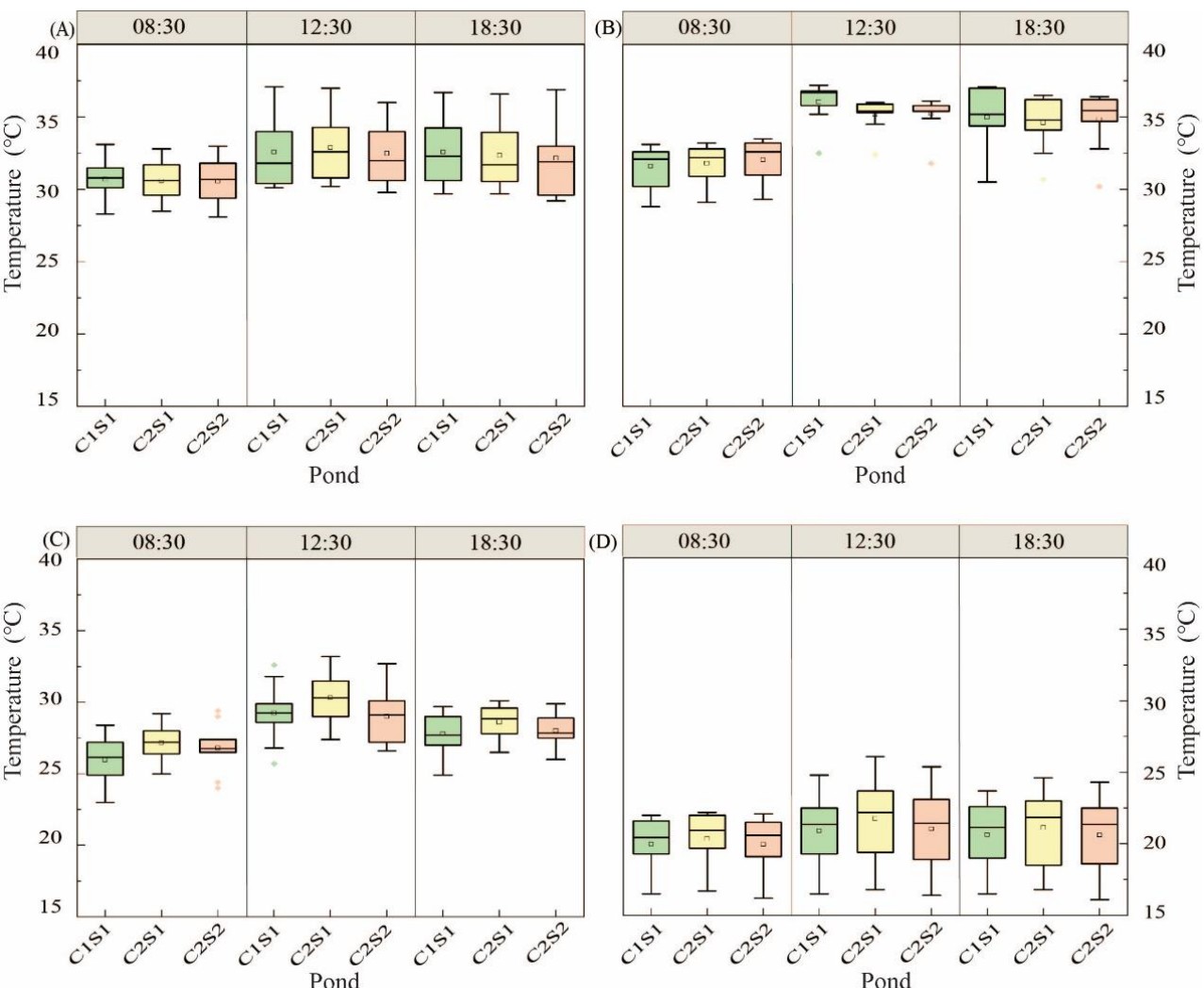

**Figure 2.** Monthly variation of water temperature in different ponds. (**A**) July; (**B**) August; (**C**) September; (**D**) October.

### 3.1.3. Change of Dissolved Oxygen in Water

For the three aquaculture ponds with different treatments, due to the consumption of one night, the DO content in the morning (8:30) was lower than that at noon (12:30 p.m.) and evening (6:30 p.m.) during the experiment period (Figure 3). However, in different months, the DO content in different ponds had different changing rules. In July, the DO content of the three treatment ponds was at a relatively low level, and there was

no significant difference in the DO content among the three ponds ($p > 0.05$, Figure 3A). In August and October, the DO content of C1S1 was significantly higher than that of the other two ponds ($p < 0.05$), but there was no significant difference between C2S1 and C2S2 (Figure 3B,D). In September, although there was no significant difference in the DO content of the three ponds in the morning ($p > 0.05$), the DO content of C2S1 at noon and evening was significantly higher than that of C1S1 and C2S2 ($p < 0.05$, Figure 3C).

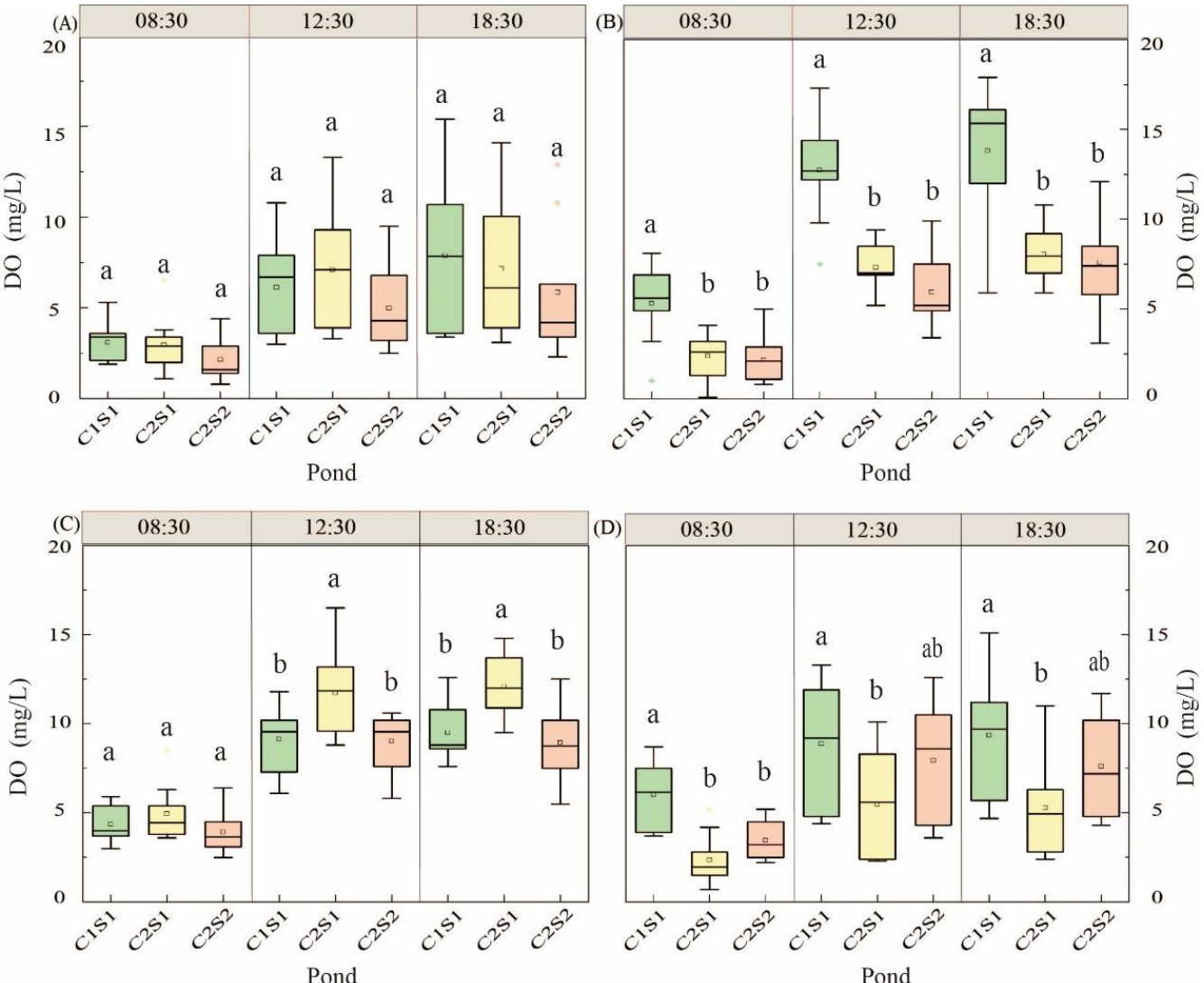

**Figure 3.** Monthly variation of water dissolved oxygen (DO) in different ponds. (**A**) July; (**B**) August; (**C**) September; (**D**) October. Values with the same letter are not significantly different (LSD post hoc test).

### 3.1.4. Ammonium-Nitrogen Change

The diurnal fluctuation of $NH_3$-N in each treatment was inconsistent and most peaks appeared in the morning and evening (Figure 4). The change of $NH_3$-N concentration in water might be related to the consumption and excreta of aquatic organisms. There was usually no significant difference in the $NH_3$-N contents of the three different treated ponds ($p > 0.05$).

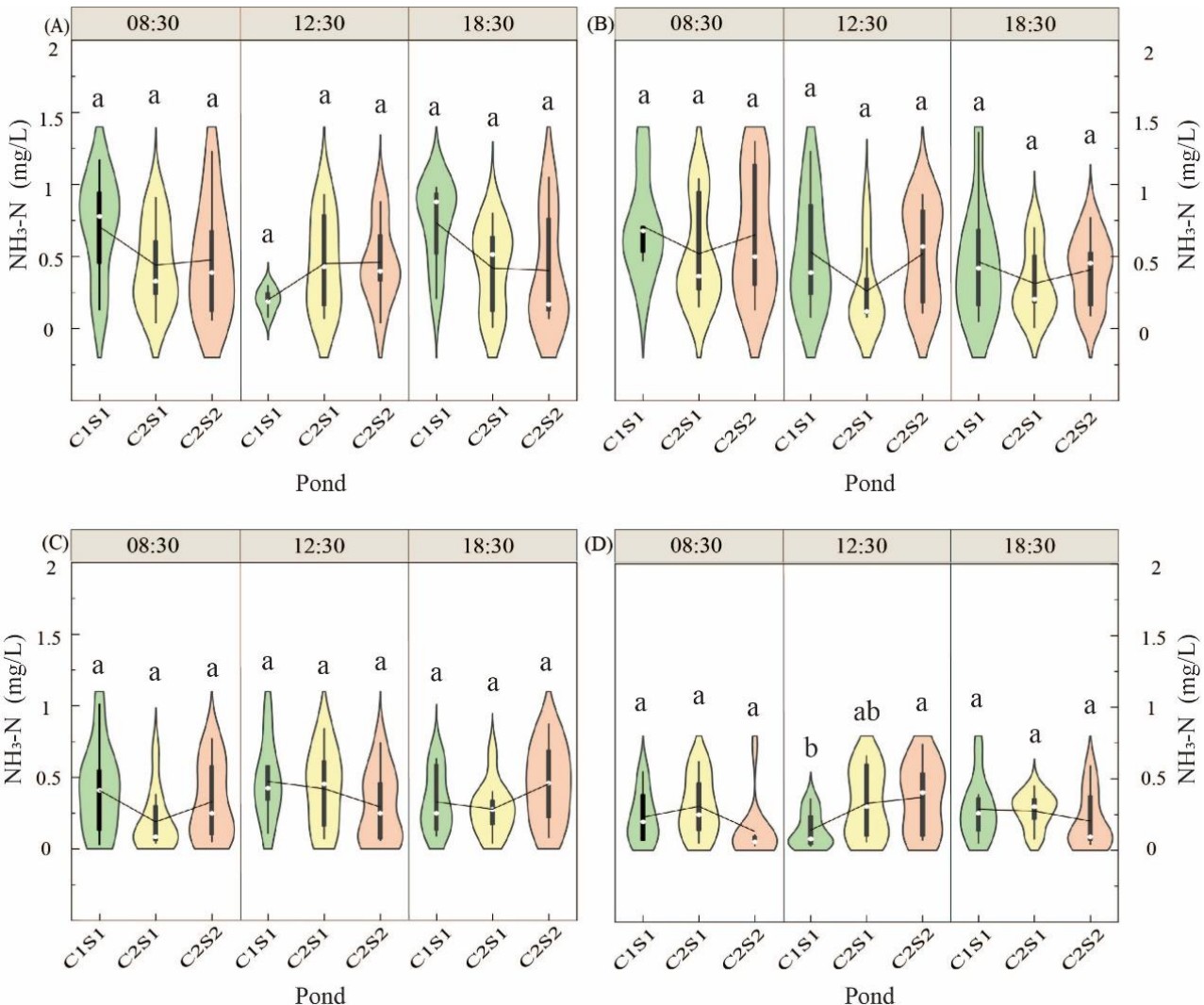

**Figure 4.** Monthly variation of water ammonium-nitrogen (NH$_3$-N) in different ponds. (**A**) July; (**B**) August; (**C**) September; (**D**) October. Values with the same letter are not significantly different (LSD post hoc test).

### 3.1.5. Orthophosphate Change

From July to October, the content of H$_3$PO$_4$ in water changed significantly (Figure 5). In July, the H$_3$PO$_4$ content of the three treatment ponds was maintained at a high level, and there was no significant difference ($p > 0.05$, Figure 5A). In August, the H$_3$PO$_4$ content of C2S2 remained at a high level, followed by C2S1, and the H$_3$PO$_4$ content of C1S1 was the lowest (Figure 5B). The H$_3$PO$_4$ concentration of C2S2 was significantly higher than that of C1S1 (Figure 5B). In September, the rules of H$_3$PO$_4$ content in the three ponds with different treatments was the same as that in August, but the H$_3$PO$_4$ content of each pond was significantly lower than that in August (Figure 5C). In October, the H$_3$PO$_4$ concentration of three ponds further decreased, and the H$_3$PO$_4$ content of C2S1 was significantly higher than that of the other two ponds ($p < 0.05$, Figure 5D).

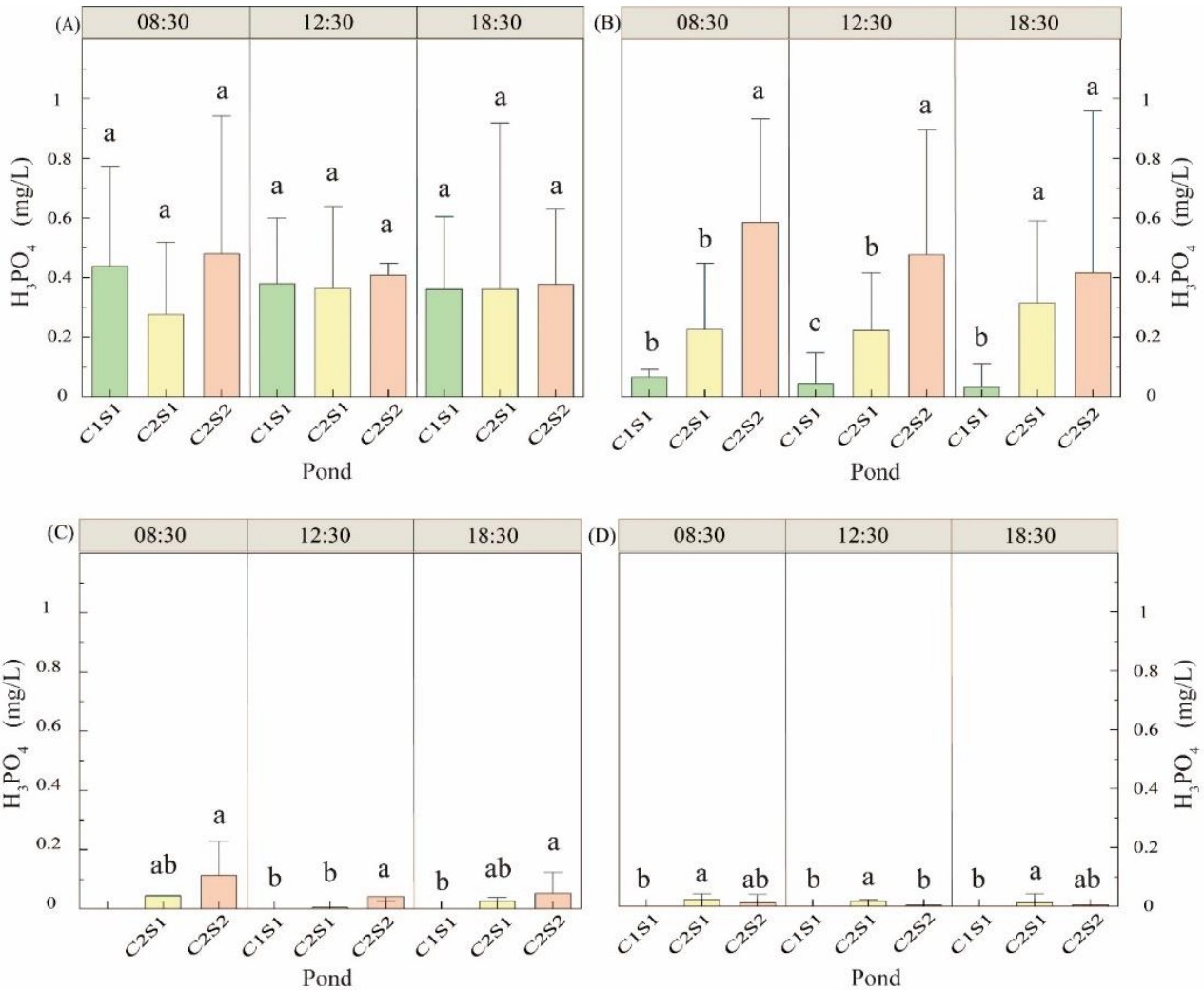

**Figure 5.** Monthly variation of water orthophosphate ($H_3PO_4$) in different pond. (**A**) July; (**B**) August; (**C**) September; (**D**) October. Values with the same letter are not significantly different (LSD post hoc test).

## 3.2. Relationship between Environmental Factors

The correlation analysis showed that $NH_3$-N and pH were significantly positively correlated ($p < 0.05$, Figure 6A). At the same time, there was a significant negative correlation between $NH_3$-N and DO ($p < 0.05$), $H_3PO_4$ and pH ($p < 0.05$). Further analysis of the relationship between surface area, waterweed coverage and environmental factors revealed that the surface area and waterweed coverage were closely related to $NH_3$-N, $H_3PO_4$ and DO, but not to pH and temperature (Figure 6B).

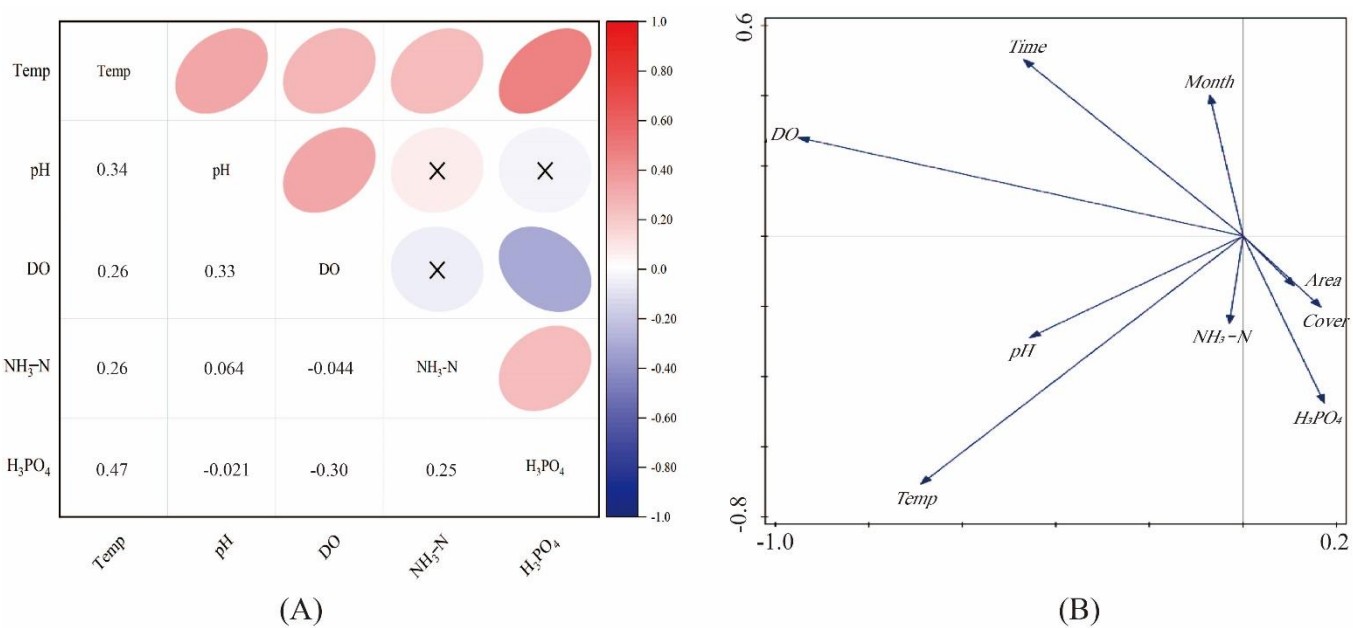

**Figure 6.** Correlation analysis (**A**) and principal component analysis (**B**) of environmental factors.

### 3.3. The Morphological Characteristics and Weight of E. sinensis

The carapace length, carapace width, carapace height and weight of *E. sinensis* in different ponds were compared (Figure 7). In the early stage of the experiment, the above indicators of Chinese mitten crabs in different ponds might not be different, but with the continuation of aquaculture activities, the above indicators of Chinese mitten crabs in different ponds began to show obvious differences in the later stage, and the above indicators in C2S2 pond were significantly higher than those in C1S1 ($p < 0.05$). The length of adult crabs in Pond C2S2 was higher than that of Pond C1S1 in July ($p < 0.05$), and there was no significant difference in August ($p > 0.05$). Starting from September, the length of *E. sinensis* in Pond C2S2 was significantly higher than that of Pond C1S1, and remained until the *E. sinensis* entered the mature period ($p < 0.05$, Figure 7A). In July and August, there was no significant difference in the width of *E.sinensis* between the three ponds ($p > 0.05$), but the width index of C2S2 was significantly higher than that of C1S1 ($p < 0.05$, Figure 7B). The height of *E. sinensis* in the three ponds had no significant difference in July ($p > 0.05$), and fluctuated in August and September (Figure 7C). The height of adult crabs in Pond C2S2 was significantly higher than that of Pond C1S1 at maturity ($p < 0.05$). Comparing the weight of *E. sinensis* in different ponds, there was no significant difference in July and August ($p > 0.05$, Figure 7D). In September, the weight of *E. sinensis* in Pond C2S2 increased rapidly, and the width and weight of *E. sinensis* in Pond C2S2 were significantly higher than the small surface area in Pond C2S1 and Pond C1S1 ($p < 0.05$).

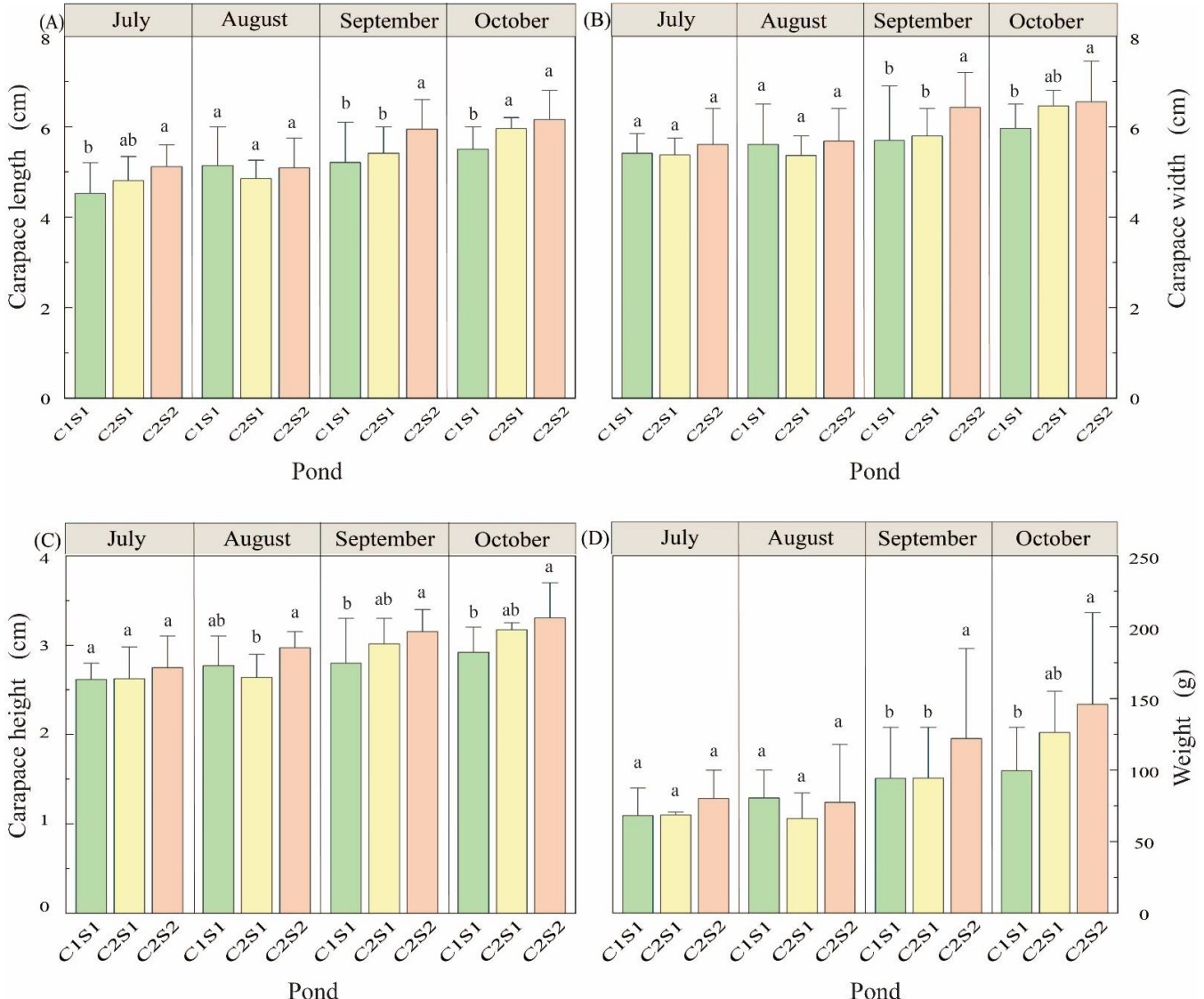

**Figure 7.** Increment dynamic of growth-related traits of *E. sinensis* in different ponds. (**A**) Carapace length; (**B**) Carapace width; (**C**) Carapace height; (**D**) Weight. Values with the same letter are not significantly different (LSD post hoc test).

*3.4. The Edible-Part Index of E. sinensis*

By measuring, dissecting, weighing and calculating the male and female crabs in each pond, the HSI, GSI, MI, G/L and other parameters of female and male crabs in different water environments were presented (Table 2). Regarding female crabs, there was no significant difference in the HSI and EPI under different treatments ($p > 0.05$). In the three ponds, the GSI of female crabs was 7.594 ± 0.671, 9.411 ± 1.753 and 7.135 ± 0.301, respectively. The GSI in Pond C2S1 was significantly higher than that in Pond C2S2. There was no significant difference in GSI of female crabs in Pond C1S1 compared with Pond C2S1 and Pond C2S2 ($p > 0.05$). Pond C2S2 was higher than that of Pond C2S1, but the difference was not significant ($p > 0.05$). All the G/L of female crabs in Pond C2S1 was significantly higher than that in Pond C2S2 ($p < 0.05$). There was no significant difference in the G/L of female crabs in Pond C1S1 compared with Ponds C2S1 and C2S2 ($p > 0.05$). The G/L of female crabs measured during the same period showed a trend of smaller surface area and larger G/L, which may be the result of factors such as accumulated temperature, nutrition and activity intensity. The MI of female crabs in Pond C2S2 was significantly higher than that in Pond C2S1 ($p < 0.05$). There was no significant difference in the MI

of adult female crabs in Pond C1S1 compared with Pond C2S1 and Pond C2S2 ($p > 0.05$). There was no significant difference in the EPI of adult female crabs in the three ponds ($p > 0.05$).

Regarding male crabs, the GSI, MI, EPI and G/L in Pond C2S2 were significantly higher than those in Pond C2S1 ($p < 0.05$). The HSI in Pond C2S1 was higher than Pond C2S2, but this difference was not significant ($p > 0.05$). From the waterweed coverage, the HSI in Pond C2S1 was significantly higher than that in Pond C1S1 ($p < 0.05$).

**Table 2.** Comparison of edible-part index of *E. sinensis* during maturation period in different ponds.

| Sex | Pond | Gonadosomatic Index (%) | Hepatosomatic Index (%) | Muscle Index (%) | Edible-Part Index (%) | G/L (%) |
|---|---|---|---|---|---|---|
| Famale | C1S1 | 7.594 ± 0.671 [ab] | 9.650 ± 0.407 [a] | 17.910 ± 2.135 [ab] | 35.154 ± 2.784 [a] | 0.787 ± 0.059 [ab] |
| | C2S1 | 9.411 ± 1.753 [a] | 9.695 ± 0.523 [a] | 15.362 ± 1.451 [b] | 34.468 ± 3.354 [a] | 0.971 ± 0.181 [a] |
| | C2S2 | 7.135 ± 0.301 [b] | 10.097 ± 1.330 [a] | 20.111 ± 1.263 [a] | 37.344 ± 2.286 [a] | 0.712 ± 0.067 [b] |
| Male | C1S1 | 2.091 ± 0.080 [a] | 9.428 ± 0.110 [b] | 19.329 ± 1.351 [ab] | 30.847 ± 1.321 [b] | 0.222 ± 0.011 [a] |
| | C2S1 | 1.345 ± 0.039 [b] | 10.070 ± 0.217 [a] | 17.887 ± 1.127 [b] | 29.302 ± 0.940 [b] | 0.134 ± 0.006 [b] |
| | C2S2 | 2.195 ± 0.491 [a] | 9.758 ± 0.276 [ab] | 21.815 ± 1.864 [a] | 33.768 ± 1.594 [a] | 0.226 ± 0.056 [a] |

Note: The different superscript beside the mean value indicates the significant difference ($p < 0.05$), while the same superscript indicates no significant difference ($p > 0.05$).

### 3.5. The Yields of E. sinensis

We compared the growth rates of *E. sinensis* from July to October (Figure 8A). In comparison with different waterweed coverages, the growth rates of *E. sinensis* in Pond C2S1 with high waterweed coverage increased over time, while the growth rates in Pond C1S1 with low waterweed coverage slowed down from July to October, which were 53% and 41%, 26%, 11% respectively. In comparison with different surface areas, the growth rates of *E. sinensis* in Pond C2S2 with the large surface area from July to August were higher than that of in Pond C2S1 with the small surface area. In the middle of September, *E. sinensis* in Pond C2S1 experienced their last molting and then grew rapidly, with a growth rate of 136%, which exceeded that in Pond C2S2.

In the final stage of the experiment, the yields of different ponds were also counted (Figure 8B). In comparison with the three ponds, C2S2 had the highest yield, reaching 0.1311 kg/m², which was more than twice that of C1S1. The yields of the C2S1 and C1S1 reached 0.1111 kg/m² and 0.0600 kg/m², respectively.

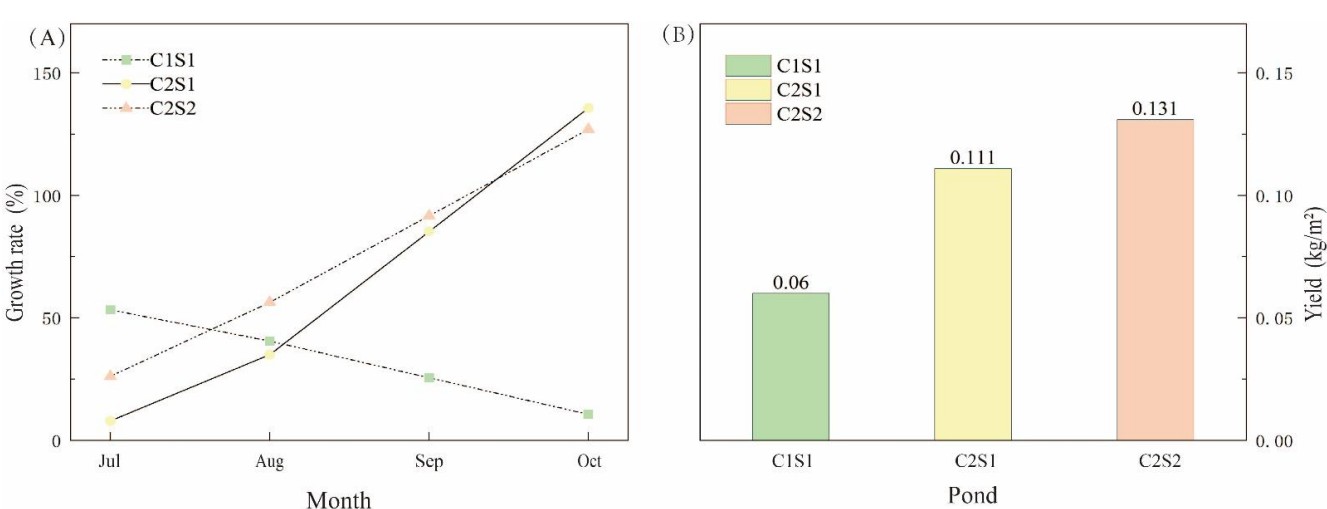

**Figure 8.** Growth-related traits of *E. sinensis* in different ponds (**A**) and yields of different ponds (**B**).

## 4. Discussion

### 4.1. The Effect of Waterweed Coverage on the Growth of Crabs

In the process of crabs culture, the waterweed coverage is the main factor affecting the ecological environment of aquaculture. In this work, *E. canadensis* was the main aquatic plant in three ponds. *E. canadensis* was a submerged plant with high quality, fast growth rate and high yield. It is wealthy in nutrients and can be used as a plant-based food for Chinese mitten crabs [13,18]. Compared to waterweed coverage ponds with low waterweed coverage ponds, the growth rate and the size of crabs in the former were better. Previous studies have also shown that the yield of aquatic grass covering 60–70% is the highest [19,20]. In fact, late August to early October is the final weight gain stage for the crabs. Yu Y. et al., found that adding a 15% protein compound feed can meet the nutritional needs of crabs, reduce culture costs, and improve water quality [21].

Under the condition of ensuring the quantity and quality of bait at the present stage, ponds with low waterweed coverage lacked plant-based food, so the growth rate of crabs and the nutrient accumulation of male crabs were significantly lower than those of high waterweed coverage ponds.

The waterweed also exerts a significant influence on regulating water quality [22,23]. Ponds with high waterweed coverage had relatively low nitrogen and phosphorus, which may be due to the role of aquatic grass. Among the three aquaculture ponds, eutrophication, which is a common problem in the lakes of China, occurred in Pond C1S1 in September. The waterweed is a good way to avoid the eutrophication of water. It can absorb excess nitrogen and phosphorus [24]. Furthermore, in the high-temperature season, waterweed can play a cooling role to prevent the damage caused by high temperature to the growth of adult crabs [25]. Peng Y et al., found that ponds with photovoltaic panels can decrease water temperature and promote the growth performance and amino acid nutrition of crabs [26].

The fast growth period of *E. canadensis* is from mid-April to early June and from late September to early November. *E. canadensis* is not resistant to high temperatures. If the water temperature reaches 30 °C or higher, its growth will be significantly weakened. It means that the leaves will turn yellow, and the top parts will wilt. In serious cases, it is easy to occur in water pollution. Therefore, to ensure the cultural environment for crabs, the waterweed should be managed in stages. In the initial stage, high waterweed coverage should be maintained to ensure healthy water quality and rich nutrition for crabs. At the same time, in addition to *E. canadensis*, it can also be matched with other aquatic plants to ensure the purification ability of water quality during high-temperature periods. From mid-July to mid-August, because of the rising temperature, the growth of *E. canadensis* is stagnant and easy to corrupt. Therefore, during the high-temperature season, we should control the growth of *E. canadensis* by cutting the grass outcropping and removing dead waterweed in time. First of all, it can prevent the death of *E. canadensis* from destroying water quality. It can then reduce oxygen consumption at night and increase dissolved oxygen. Finally, it can make rhizomes strong of *E. canadensis* and survive until the end of aquaculture.

### 4.2. The Effect of the Surface Area of the Ponds on the Growth of Crabs

Wang et al., showed that pond size was closely related to crab yield [27]. In the process of crab culture, carbs in the large surface areas grew faster than in the small surface areas, and crabs were larger in size and had a higher MI. In the case of a certain water depth, the large surface area guaranteed the survival and activity space of crabs, and correspondingly increased the feeding range. In this way, the probability of obtaining bait per unit area is high, which can better promote growth. According to Allen KO's research, the increased cultural density often causes density stress, and increases the energy consumption in order to adapt to this environment, resulting in a significant decrease in the growth rate, size and survival rate of the breeding species [8]. Li's research also showed that high cultural density destroyed the stability of the community structure of crab ponds and adversely affected the water environment [28]. Wang Xing et al. [29] studied the effects of different

feeding densities on the growth performance and antioxidant capacity of *E. sinensis*, and they found that the optimal stocking density of *E. sinensis* in the rice crab culture system is 0.4 individuals/m². This is consistent with this research.

The gonad weight of female crabs is the main selling point on the consumer market, which is commonly known as crab roe. The GSI of female crabs in ponds with the small surface area was significantly higher than that in ponds with the large surface area. The reason may be that the competition for bait increased while the growth space of crabs was small. At the same time, when the crab molt, the risk of being eaten by other crabs increased, which promoted early molting and energy storage.

For the culture of *E. sinensis*, sufficient living space should be provided, and ponds with the large surface area should be selected as much as possible. If the surface area is less than 20 acres, culture density can be appropriately reduced. A reasonable culture density will prevent the water environment from significant contamination, ensure continuous production, and increase the economic benefits of breeding personnel.

*4.3. The Effect of Other Factors on the Growth of Crabs*

The continuously high temperature in July and August may be the main reason for the slow growth and death of crabs. Water temperature has an important effect on the reproduction, growth and feeding of aquatic animals. Each aquatic animal shares a suitable water temperature range and an optimum water temperature for its growth. In most cases, within the suitable range of water temperature, the food intake and the growth rate of aquatic animals gradually increase with the rising temperature. The longer the optimum water temperature lasts, the faster individual aquatic animals grow [30]. After the crabs molted once in mid-July, the high-temperature period was followed. At this time, the pond temperature was relatively high, about 28 °C to 38 °C. When the temperature reached 35 °C, the crab basically entered the dormancy state and stopped eating. Moreover, the continuously high temperature of 3 to 4 days can easily induce disease and cause the death of the crabs [31]. On the other hand, affected by the continuously high temperature in July and August, aquatic plants decayed. This made the water quality worse and further affected the growth and survival of adult crabs.

The factors that influence crab yields are complex. The growth of adult crabs is controlled by genes, on the other hand, it is affected environment. The external environment affects the molting time of adult crabs and the weight gain after molting [32]. Due to the long period of culture, the water is complex and changeable, which is greatly affected by the weather. Therefore, in the process of *E. sinensis* culture, we should pay great attention to water quality monitoring and timely adjust the abnormal water quality.

In addition, the brand will also affect the production of *E. sinensis*. Xue J's research believed that the shell shape of local crabs need six months to stabilize and reflect their origin [33], which was of great significance for protecting the original habitat of crabs.

**5. Conclusions**

In order to understand the effect of water area and waterweed coverage on the growth of *E. sinensis* cultured in ponds, the water temperature, pH, dissolved oxygen, ammonia nitrogen, phosphate, and the carapace length, width, height and weight of *E. sinensis* were measured in different ponds with different coverage and different areas in July, August, September and October. The results showed that different waterweed coverage and pond area had significant effects on the growth, yield and quality of *E. sinensis*. For better economic benefits, it is recommended to select ponds with high water and grass coverage and large surface areas when raising *E. sinensis*.

**Author Contributions:** Conceptualization, Y.Z. and Z.X.; methodology, J.P. and Y.D.; software, Y.T.; J.C., Z.H., Y.D. and T.H.; formal analysis, Y.T. and Z.H.; investigation, Y.D.; resources, J.P.; data curation, M.D.; writing—original draft preparation, J.P.; writing—review and editing, Y.T.; visualization, J.D.; supervision, Z.X.; project administration, Z.X.; funding acquisition, Z.X. and M.D. Y.T. and J.P. are the co-first authors. All authors have read and agreed to the published version of the manuscript.

**Funding:** This study was supported by the National Natural Science Foundation of China (U20A20118), Natural Science Foundation of Hunan Province (2022JJ50264), and Research Foundation of Education Bureau of Hunan Province (21B0711).

**Institutional Review Board Statement:** The study was conducted according to the guidelines of the Declaration of Helsinki, and approved by the Ethics Committee of Animal Care and Use Committee of the Central South University of Forestry and Technology. All operations were carried out with field permit no. 2014BAC09B03-02.

**Informed Consent Statement:** Informed consent was obtained from all subjects involved in the study.

**Data Availability Statement:** Data are contained within the article.

**Acknowledgments:** This study would like to thank Zhang Qing who provide breeding places for the study.

**Conflicts of Interest:** The authors declare that they have no conflict of interest.

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
