# Peer review of "Effect of Water Area and Waterweed Coverage on the Growth of Pond-Reared Eriocheir sinensis"

_fishes, doi:10.3390/fishes7050282_

Round 1

Reviewer 1 Report

In general, this is a well-written manuscript,  so I recommend that this manuscript could be accepted with some minor revisions. Few comments to the authors were showed as follows:

-Line 18: The authors have “P<0.05 “, it must be written in lower case “p”. It should be changed throughout the document.

-Line 147:  The authors have “… Primary …analysis…. “ . it must be  “Principal componente analysis”

-Table1: Regarding to statistical analysis the authors should give information about p value obtained for each one of the tests. 

-Fig 3, 4, 5, 7:  Legend: the authors must put information about the letters “a”, “b”,”c”. I suggest the following: values with the same letter are not significantly different (LSD  post hoc test )

-Line217 and 218: By looking at the Fig 6 A it seems that the correlation between NH3-N and DO is near zero. I suggest put the correlation value on the text.

-Regarding PCA, can the authors explain How they considerer a qualtitativa variable “mouth” in a statistical techique that only alows quantitative variables?

Lines 225, 233, 234, …, 242:”…E. sinensis…”

Lines 252 until 274: I don´t understand the “% “after de standard deviation. This information is about mean and standard deviation. These values are not percentages. Please correct full text.

Section 3.5: The name of the sepcies shoulb be in italic.

Fig 8:”…E. sinensis…”

Lines 323, 329, 331, …336:”…E. sinensis…”

Line339: showed

Line 381 :”…E. sinensis…”

Reviewer 2 Report

Review of ‘Effect of Water Area and Waterweed Coverage on the Growth of Pond-Reared Eriocheir Sinensis by Tang Yongcheng, Peng Jiao, Chen Jiahao, Zhao Yunlin, Ding Yi, Dai Jingyi, Hu Zhiyuan, Huang Tian, Dong Meng and Xu Zhenggang.

The authors conducted an interesting study to reveal the effects of water area and waterweed coverage on the growth performance of the Chinese mitten crab (Eriocheir sinensis). The authors described water quality parameters and their dynamics over the study period and related environmental fluctuations to growth rates in the crabs. They concluded that ponds with high waterweed coverage and large area provide the highest yield of the final product. After some minor revisions this paper can be recommended for publication in FISHES.

General remark.

Figure captions. The authors should specify what different letters (a, b, c…) mean.

The Latin names must be italicized throughout the text.

The authors should update the discussion with some recent papers:

Pang, Y.; Niu, C.; Wu, L.; Song, Y.; Song, X.; Shi, A.-y.; Shi, X.; Wu, Z.-w.; Tang, B.; Yang, X.; Cheng, Y. Comprehensive Utilization of Land Resources for Photovoltaic Power Generation to Culture Chinese Mitten Crab (Eriocheir sinensis): Growth Performance, Nutritional Composition and Tissue Color. Fishes 2022, 7, 207.

Yu, Y.; Wan, J.; Liang, X.; Wang, Y.; Liu, X.; Mei, J.; Sun, N.; Li, X. Effects of Protein Level on the Production and Growth Performance of Juvenile Chinese Mitten Crab (Eriocheir sinensis) and Environmental Parameters in Paddy Fields. Water 2022, 14, 1941.

Wang, X., Yao, Q., Lei, Xy. et al. (2022) Effects of different stocking densities on the growth performance and antioxidant capacity of Chinese mitten crab (Eriocheir sinensis) in rice crab culture system. Aquacult Int 30, 883–898

J. Xue, H. Liu, T. Jiang, X. Chen & J. Yang (2022) Shape variation in the carapace of Chinese mitten crabs (Eriocheir sinensis H. Milne Edwards, 1853) in Yangcheng Lake during the year-long culture period, The European Zoological Journal, 89:1, 217-228

Abstract.

Pg 1 Ln 14: Suggest changing ‘pond’ to ‘ponds’

Pg 1 Ln 21: Suggest changing ‘significantly  negatively  correlation’ to ‘a significant negative correlation’

Pg 1 Ln 28: Suggest changing ‘benefits was believed’ to ‘benefits’

Introduction

Pg 1 Ln 32: Suggest changing ‘were  rich  in  protein’ to ‘are  rich  in  protein’

Pg 1 Ln 33: Suggest changing ‘HDA’ to ‘DHA’

Pg 1 Ln 33: Suggest changing ‘which  were  indispensable’ to ‘which  are  indispensable’

Pg 1 Ln 36: Suggest changing ‘China was’ to ‘China is’

Pg 2 Ln 46: Suggest changing ‘were cultivated to be button size and then to be market size’ to ‘are cultivated to a button size and then to a marketable size’

Please, indicate the marketable size for this species.

Pg 2 Ln 50: Suggest changing ‘were  employed  and  crabs  were’ to ‘aremployed  and  crabs  are’

Pg 2 Ln 66: Suggest changing ‘factor for the culture’ to ‘factor inhe culture’

Pg 2 Ln 69: Suggest changing ‘to high density environment’ to ‘to a high-density environment’

Pg 2 Ln 70: Suggest changing ‘in the decrease of’ to ‘in a decrease in’

Pg 2 Ln 74: Suggest changing ‘area’ to ‘areas’

Pg 2 Ln 75: Suggest changing ‘environment of’ to ‘environment for’

Pg 2 Ln 77: Suggest changing ‘area’ to ‘areas’

Materials and Methods

Pg 2 Ln 83: Suggest changing ‘in 2016’ to ‘2016’

Pg 3 Ln 102: Suggest changing ‘as follow’ to ‘as follows’

Pg 3 Ln 121: Suggest changing ‘October in 2016’ to ‘October 2016’

Pg 3 Ln 123: Suggest changing ‘Crabs’ to ‘The crabs were’

Pg 4 Ln 147: Were the data normally distributed to applying the parametric analysis?

Pg 4 Ln 147: As I know, PCA means ‘Principal component analysis‘.

Results

Pg 4 Ln 161: Suggest changing ‘pond’ to ‘ponds’

Pg 4 Ln 164: Suggest changing ‘reached the peak’ to ‘reached a peak’

Pg 4 Ln 172: Suggest changing ‘difference for’ to ‘difference in’

Pg 5 Ln 175: Suggest changing ‘pond’ to ‘ponds’

Pg 6 Ln 190: Suggest changing ‘pond’ to ‘ponds’

Pg 6 Ln 193: Suggest changing ‘most of  peak’ to ‘most peaks’

Pg 6 Ln 196: Suggest changing ‘In most of the time, there was no significant difference for’ to ‘Usually, there was no significant difference in’

Pg 7 Ln 199: Suggest changing ‘pond’ to ‘ponds’

Pg 8 Ln 217: Suggest changing ‘significantly  negatively’ to ‘a significant  negative’

Pg 9 Ln 234: Suggest changing ‘entering the mature period’ to ‘enteredhe mature period’

Pg 9 Ln 235: Please explain what do mean ‘the width index’.

Pg 9 Ln 237: Please explain what do mean ‘the height index’.

Pg 9 Ln 243: Suggest changing ‘were significant. Higher’ to ‘were significantly higher’

Pg 10 Ln 250-278: I suggest shortening this section because the text repeats information presented in Table 2.

Pg 10 Ln 289: Suggest changing ‘August was’ to ‘August were’

Discussion

Pg 11 Ln 306: Suggest changing ‘Compared high’ to ‘Compared to’

Pg 11 Ln 310: Suggest changing ‘stage of’ to ‘stage for’

Pg 11 Ln 320: Suggest changing ‘temperature on’ to ‘temperature to’

Pg 11 Ln 328: Suggest changing ‘nutrition of’ to ‘nutrition for’

Pg 11 Ln 330: Suggest changing ‘ability for’ to ‘ability of’

Pg 12 Ln 347: Suggest changing ‘specie’ to ‘species’

Conclusions

Pg 13 Ln 387: Suggest changing ‘different area’ to ‘different areas’
